# Does the Immunohistochemical Expression of CD44, MMP-2, and MMP-9 in Association with the Histopathological Subtype of Renal Cell Carcinoma Affect the Survival of Patients with Renal Cancer?

**DOI:** 10.3390/cancers15041202

**Published:** 2023-02-14

**Authors:** Magdalena Chrabańska, Magdalena Rynkiewicz, Paweł Kiczmer, Bogna Drozdzowska

**Affiliations:** Department and Chair of Pathomorphology, Faculty of Medical Sciences in Zabrze, Medical University of Silesia, 40-055 Katowice, Poland

**Keywords:** CD-44, MMP-2, MMP-9, renal cell carcinoma, immunohistochemistry

## Abstract

**Simple Summary:**

The clinical outcomes of renal cell carcinoma (RCC) differ widely, indicating the need for accurate prognostic parameters. Because cancer stem cells and matrix metalloproteinases play a key role in carcinogenesis, CD44, MMP-2, and MMP-9 may be the potential prognosticators for RCC. The aim of our study was to analyse whether the immunohistochemical expression of these molecules in association with the histopathological RCC subtype affects patients’ survival. Significant differences existed in the degree of MMP expression between clear-cell RCC and non-clear cell RCC cases, which suggests different tumorigenic mechanisms between these subtypes. On multivariate analysis, only the histopathological subtype of clear cell RCC and CD44 expression were independent risk factors for patient death. Thus, CD44 seems to be an independent factor of poor outcomes in patients with RCC regardless of its subtype. It may be useful in the search for new therapeutic methods and in predicting the prognosis of patients with RCC.

**Abstract:**

CD44, MMP-2, and MMP-9 are new potential molecular prognostic markers in renal cell carcinoma (RCC). The aim of the study was to analyze whether the expression of CD44, MMP-2, and MMP-9 in association with the histopathological subtype of RCC affects the survival of patients with renal cancer. The study population included 243 clear cell RCC (ccRCC) and 59 non-ccRCC cases. A total of 302 tumors were examined for CD44, MMP2, and MMP9 expression by immunohistochemistry. The expression levels of the proteins were scored by semi-quantitative methods, and the correlation with overall patient survival was verified. We found no significant differences in CD44 expression levels between cc-RCC and non-ccRCC cases; however, significant differences existed in the degree of MMP-2 and MMP-9 expression between cc-RCC and non-ccRCC cases. There was significantly higher MMP expression in non-ccRCC than in ccRCC cases. Univariate Cox regression analysis showed that increased CD44 expression and histopathological subtype of ccRCC were predictors of shorter overall survival. Moreover, MMP-2 overexpression slightly reduced the risk of patient death, while MMP-9 expression did not show an association with patients’ survival. However, on multivariate analysis, only the histopathological subtypes of ccRCC and CD44 expression were independent risk factors for patient death.

## 1. Introduction

Renal cell carcinoma (RCC) is the seventh most common form of neoplasm in the developed world, accounting for approximately 2% of global cancer diagnoses and deaths and projected to increase in burden worldwide [1,2]. RCC is comprised of several histological cell types. Every subtype arises from a variety of specialized cells located along the length of the nephron. Both clear-cell (ccRCC) and papillary (pRCC) RCCs are believed to arise from the epithelium of the proximal tubule, while chromophobe RCC (chRCC) is thought to arise from the distal nephron, probably from the epithelium of the collecting tubule [3]. According to the fourth edition of the World Health Organization (WHO) classification of urogenital tumors, ccRCC represents 65–70% of RCC, pRCC accounts for 18.5% of RCC, and chRCC constitutes 5–7% of RCC [4]. Each type has differences in genetics, biology and behaviour. In the case of RCC, prognostic factors can be sub-classified into anatomical, histological, clinical and molecular factors. Generally, anatomical and histological prognostic factors possess a higher level of evidence than clinical or molecular prognostic factors. The traditional anatomical prognostic factor includes the TNM staging system, which incorporates several prognostic characteristics, such as the size of the tumor, invasion of different kidney structures, and spread to the regional lymph nodes or distant locations. Histological prognostic factors include RCC subtype, tumor grade, presence of sarcomatoid or rhabdoid transformations, microvascular invasion, and tumor necrosis. Additionally, some clinical prognostic factors have been investigated, such as performance status, presenting symptoms, paraneoplastic syndromes, and different laboratory tests [5]. Unfortunately, these parameters are not well-founded for predicting the biological behaviour of RCC. The recognition of new molecular markers can improve the comprehension of renal cancer biology and lead to the evolution of revolutionary targeted therapies. One of these molecular prognostic markers could be CD44, which is a compelling marker for cancer stem cells (CSCs) of many solid malignancies, including RCC [6,7]. CSCs are tightly involved in all stages of carcinogenesis, such as initiation, promotion, progression, metastasis and the eventual recurrence of cancer; thus, eliminating CSCs is a crucial step in cancer therapy [6,7,8,9]. The CD44 protein is engaged in many cellular functions critical to cancer progression and metastasis, including migration and cell–cell and cell–matrix adhesion. Furthermore, CD44 binds to the extracellular matrix and acts as a platform for growth factors and matrix metalloproteinases (MMPs) [6,10,11], which are other prognostic molecular markers for RCC. Of these indicators, MMP-2 and MMP-9, in particular, which are responsible for degrading the major ingredient of the basement membrane, are indicated. These two collagenases are commonly related to the malignant phenotype of tumor cells because they intensify tumorigenicity. Along with their invasive function, MMPs are also related to cell proliferation and angiogenesis [12,13,14].

Currently, a limited number of prior studies have investigated the immunohistochemical expression of CD44, MMP-2, and MMP-9 in RCC. The vast majority of studies have explored these prognostic markers only in the context of ccRCC. Moreover, not all of these studies were recorded using immunohistochemistry; some of them used other techniques. To our best knowledge, this is the largest study investigating the prognostic role of CD44, MMP-2, and MMP-9 in different histopathological subtypes of renal cancer.

Hence, the aim of this study was to analyse whether the immunohistochemical expression of CD44, MMP-2, and MMP-9 in association with the histopathological subtype of RCC affects the survival of patients with renal cancer.

## 2. Materials and Methods

The study was conducted in accordance with the Declaration of Helsinki and approved by the Institutional Review Board of the Medical University of Silesia, Katowice, Poland (protocol code KNW/0022/KB/228/19). Patient data were kept fully anonymous in all steps. 

In reporting this study, the “Strengthening the Reporting of Observational Studies in Epidemiology” (STROBE) guidelines were used [15].

### 2.1. Patients and Tumor Samples

All 302 patients with a diagnosis of RCC, including 243 ccRCC and 59 non-ccRCC (41 pRCC and 18 chRCC), were involved in this study. Patients with other histological types of renal carcinoma were not included. All patients underwent partial or radical nephrectomy for sporadic RCC between June 2015 and October 2020. All surgical specimens were handled according to the current guidelines of the ISUP and the WHO for specimen handling, sampling, and reporting [16,17]. The tissue specimens were formalin-fixed and paraffin-embedded. The hematoxylin and eosin-stained slides were reviewed by two independent pathologists. All sections were then assessed for tumor size, tumor (T) stage, WHO/ISUP grade, presence and percentage of necrosis, sarcomatoid and rhabdoid differentiation, small vessel lymphovascular invasion, fibrous renal capsule invasion, perinephric fat invasion, renal sinus fat, and vascular invasion of renal sinus vessels. Follow-up details contained the date of nephrectomy, survival status, date of death, and/or date of last follow-up.

### 2.2. Immunohistochemical Staining and Its Evaluation

For each tumor, a representative slide and the corresponding paraffin block were selected. The immunohistochemical staining was performed in an automated immunostainer in agreement with the manufacturer’s requirement (Table 1) and the same methodology which was used and described in detail in our previous study [18].

The staining results were independently examined by two pathologists, who were completely blinded for medical and pathological data of patients. Semi-quantitative analysis was performed to evaluate the CD44, MMP-2, and MMP-9 expression. The modified Allred et al. method was used to evaluate both the intensity and the proportion of immunohistochemical staining [18,19]. The intensity scores ranged from negative to strong, as follows: 0 = negative, 1 = weak, 2 = moderate, and 3 = strong. The proportion scores ranged from 0 to 5 and were categorized according to the positive tumor cells as follows: 0 = no staining, 1 = up to 1/100 positive cells, 2 = 1/100 to 1/10 positive cells, 3 = 1/10 to 1/3 positive cells, 4 = 1/3 to 2/3 positive cells, 5 => 2/3 positive cells. To calculate the total immunohistochemical score, the proportion and intensity scores were multiplied for each specimen (ranging from 0 to 15) [20]. Then, overall immunohistochemical scores were classified into three groups as follows: 0–5 as Group 1 (low expression), 6–10 as Group 2 (moderate expression), and 11–15 as Group 3 (high expression).

### 2.3. Statistical Analysis

All analyses were performed using R Language in the Rstudio Environment. Quantitative data were presented as numbers, case percentages, mean, and standard deviation (SD). Pearson’s chi-squared (Chi^2^) test was used to find differences between qualitative variables. Correlation analysis was conducted using Spearman’s rank coefficient for quantitative variables or Kendall’s Tau correlation coefficient for semi-quantitative ones. Survival analysis was performed using the Kaplan–Meier estimator with the Gehan–Wilcoxon test for two groups and with Mantel correction for multiple groups to determine differences between groups. A uni- and multivariate Cox’s proportional hazard model was prepared to analyze the effect of examined variables on patients’ outcomes. The effect size was calculated for each test using Cramer’s V for Chi^2^ tests. The power of each test was calculated using G*Power software or the pwr package for R language in Rstudio software. Statistical significance was set at *p* < 0.05.

## 3. Results

### 3.1. Clinicopathological Data

The sample population included a total of 302 RCC tumors: 243 ccRCC and 59 non-ccRCC. The group of non-ccRCC consisted of 41 pRCC and 18 chRCC; however, due to the small number of cases with these types of cancers, we decided to combine them into one group. In the case of non-ccRCC, nephron-sparing surgery (NSS) was performed statistically more often than in the group of non-ccRCC, probably because these tumors were significantly smaller than cc-RCC. Moreover, compared to ccRCC, non-ccRCC (specifically pRCC) were characterized by a significantly lower WHO/ISUP nuclear grade, which is one of the most important histological prognostic factors for renal cancer. This relationship may then account for a significant difference in survival between cc-RCC and non-ccRCC patients. The comparison of the clinicopathological parameters of RCC patients is summarized in Table 2. 

### 3.2. Immunohistochemical Staining

All 302 investigated RCC cases were analyzed immunohistochemically for CD44, MMP-2, and MMP-9 expression. In the case of ccRCC, 99 (40.74%) samples were negative for CD44, 218 (89.71%) for MMP-2, and 194 (79.84%) for MMP-9. Regarding non-ccRCC cases, of all samples stained for CD44, MMP-2 and MMP-9, 11 (18.64%), 9 (15.25%) and 24 (40.68%) samples did not show any staining, respectively. In our study, no significant difference in CD44 expression was observed between cc-RCC and non-ccRCC cases; however, significant differences existed in the degree of MMP-2 and MMP-9 expression between cc-RCC and non-ccRCC cases. There was significantly higher MMP expression in non-ccRCC than in ccRCC cases. The comparison of the immunohistochemical distributions of CD44, MMP-2, and MMP-9 of RCC cases based on the overall immunohistochemical score is presented in Table 3. 

### 3.3. Prognostic Value of CD44, MMP-2, and MMP-9 Expression in Association with the Histopathological Subtype of Renal Cell Carcinoma

The mean duration of follow-up was 48.1 months (SD = 24.8), with a median of 49.3 months (interquartile range = 28.9 to 69.2 months). All deaths were observed during the 85 months after surgery. During this time, death was documented in 79 (32.51%) ccRCC patients and in 6 (10.17%) non-ccRCC patients. All deaths in the group of non-ccRCC concerned only patients with diagnosed pRCC.

Univariate Cox analysis was performed to assess the clinical significance of evaluated parameters that might influence overall survival (OS) in patients with RCC. As summarized in Table 4, increased CD44 expression was a predictor of shorter OS–patients with increased CD44 expression had a significantly greater risk of death than patients with low CD44 expression. Furthermore, the results of the statistical analysis demonstrated that MMP-2 slightly reduced the risk of patient death, while MMP-9 expression did not show an association with patients’ survival. In addition, univariate Cox regression analysis showed that the histopathological subtype of RCC was associated with patient prognosis–patients with ccRCC had a significantly higher risk of death than patients with non-ccRCC (Table 4, Figure 1).

To discover the independent prognostic factors, multivariate analysis was carried out using Cox’s proportional hazards model. This analysis showed that both histopathological subtypes of ccRCC and CD44 expression were independent risk factors for patient death (Table 5). However, no interaction between CD44 expression and RCC subtype was observed in our study.

## 4. Discussion

Despite recent advances in the diagnosis and treatment of RCC, it is still a tumor of unpredictable presentation and clinical outcome. It is generally agreed that the conventional evaluation of prognosis for RCC patients depends predominantly on histopathological characteristics, nuclear grade, and the TNM staging system [21,22,23]. However, in many patients with RCC, these prognostic parameters are not sufficient to predict the long-term clinical course of the disease. Thus, determining new prognostic factors would help in following up with patients and in directing further therapy. Recently, markers such as CD44, MMP-2, and MMP-9 have been studied in order to improve the probability of predicting the prognosis of RCC. Therefore, in this research, we answered the question of whether the immunohistochemical expression of CD44, MMP-2, and MMP-9 in association with the histopathological subtype of RCC affects the survival of patients with renal cancer. 

CD44 is an important cancer stem cell marker and a poor prognostic marker in various malignancies. It helps in various steps which are fundamental in the extravasation and migration of neoplastic cells [22]. CD44 is an adhesion molecule that binds to the extracellular matrix and is implicated in cancer cell migration, invasion and metastasis. High CD44 expression is correlated with CSC-like phenotypes; therefore, blockade of CD44 may prevent the progression of cancer and slow the process of metastasis, recurrence, and resistance to chemotherapeutic agents [10]. 

Matrix metalloproteinases are a family of zinc-dependent proteases, and MMP-2 and MMP-9 are major members of the MMP family. They are mainly secreted by tumor cells and stromal cells and, after activation, facilitate tumor expansion and the promotion of metastasis by mediating the degradation of basement membrane and connective tissue barriers [24]. MMP-2 and MMP-9 have also been implicated in cancer development and progression through their functions in cell apoptosis, proliferation, and angiogenesis [25]. 

The vast majority of studies have explored CD44, MMP-2, and MMP-9 only in the context of ccRCC. For patients with non-clear-cell histology, there is little consensus. Lee et al. [26], who examined a total of 107 ccRCCs and 32 non-ccRCCs, showed in univariate and multivariate analyses that increased CD44 expression was an independent predictor of shorter OS only in the ccRCC group. In the non-ccRCC group, univariate analysis showed a positive correlation between CD44 expression and shorter OS, but multivariate analysis did not reach a statistically significant result. A similar observation has been made by Zanjani et al. [10], who evaluated CD44 expression in 136 ccRCCs and 70 non-ccRCCs. Unfortunately, other studies examining the role of CD44 as a prognostic marker in kidney cancer evaluated it only in the case of the ccRCC. In the study of Jeong et al. [11] the multivariate analysis showed that CD44 expression was an independent risk factor predicting recurrence-free survival, disease-specific survival, and overall survival in patients with ccRCC. Additionally, Gayyed et al. [22] found that CD44 overexpression was significantly related to the overall survival of ccRCC patients. In contrast to these results, Costa et al. [21] reported that CD44 tissue expression was not an independent predictor of disease-specific survival or progression-free survival. 

Regarding MMPs, there are only a few studies describing their impact on RCC, and many of their results are contradictory. Tissue MMP-2 and MMP-9 were found to be overexpressed in renal tumors but more frequently expressed in non-ccRCC compared with clear-cell tumors [27,28]. Additionally, in our study, we found significantly higher MMP expression in non-ccRCC than in ccRCC cases. In the research of Kallakury et al. [27], on univariate analysis, only the increased expression of MMP-9 related to a shortened RCC patient survival. However, on multivariate analysis, overexpression of any of the investigated markers did not reach statistical significance in their study. Lee et al. [26] reported that MMP-2 and MMP-9 expression did not show an association with survival in the univariate or multivariate analyses in either the ccRCC or non-ccRCC patients.

As there are not many studies examining the role of CD44, MMP-2, and MMP-9 in the carcinogenesis of RCC, especially in non-ccRCC, further research in a large cohort of patients is needed to understand the mechanism of RCC tumorigenesis and develop therapeutic stratification.

## 5. Limitations

The essential limitations of our study were the relatively small number of non-ccRCC tumors included in this study and the large difference in the size of the ccRCC and non-ccRCC groups. This was due to the rarity of papillary and chromophobe RCC when compared to ccRCC. The next limitation concerned the limited number of prior research comparing these two groups of neoplasms. There are only a few studies describing the role of MMPs in RCC carcinogenesis, and additionally, not all of them discovered these proteins using immunohistochemistry.

## 6. Conclusions

In summary, the present study showed no significant difference in CD44 expression between cc-RCC and non-ccRCC cases; however, there was significantly higher MMP-2 and MMP-9 expression in non-ccRCC than in ccRCC tumors, which may suggest the presence of different tumorigenic mechanisms between these RCC subtypes. Moreover, in univariate and multivariate analyses, both histopathological subtype of ccRCC and CD44 overexpression were independent risk factors for RCC patient death. Thus, CD44 seems to be an independent factor of poor outcomes in patients with RCC regardless of its subtype. Therefore, this finding indicates CD44 as a possible predicting factor for the outcome of RCC patients and a possible target for new therapeutic approaches. 

## Figures and Tables

**Figure 1 cancers-15-01202-f001:**
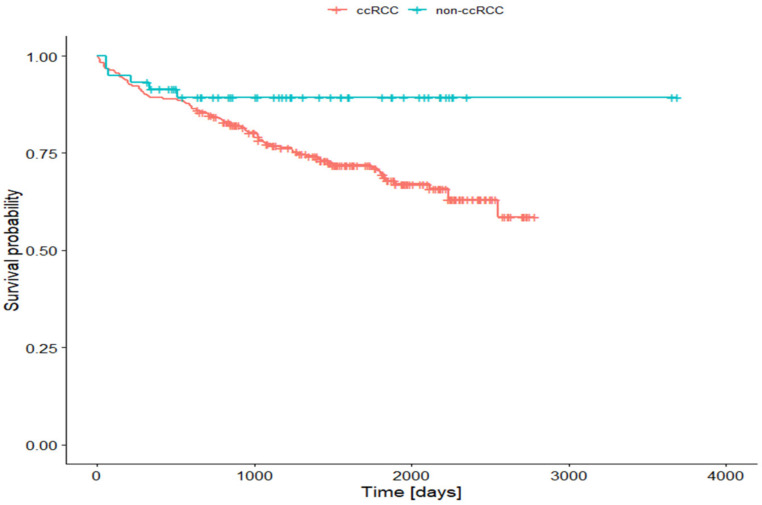
Kaplan–Meier survival curves according to the histopathological subtype of renal cell carcinoma. Patients with clear cell renal cell carcinoma had a significantly higher risk of death than patients with non-clear-cell renal cell carcinoma.

**Table 1 cancers-15-01202-t001:** Characteristics of the antibodies used for immunohistochemistry.

Antibody	Clone	Source	Dilution	Incubation (min)
CD44	Monoclonal	MRQ-13, Cell Marque, Rocklin, CA, USA	1:300	20
MMP-2	Monoclonal	CA-4001, Zeta Corporation, Arcadia, CA, USA	1:50	60
MMP-9	Monoclonal	EP127, Bio SB, Goleta, CA, USA	1:100	60

**Table 2 cancers-15-01202-t002:** Comparison of the clinicopathological characteristics of clear-cell and non-clear-cell renal cell carcinoma cases.

	Clear Cell RCC	Non-Clear Cell RCC	Power/Effect Size	*p*-Value
Number of tumor samples (*n*)	243	59		
Age, years (mean ± SD)	63.7 ± 10.2	63.2 ± 9.1	0.166/0.08	0.5707 ^c^
Sex (*n* (%))				
Female	89 (36.62%)	16 (38.89%)	0.23/0.07	0.2436
Male	154 (63.37%)	43 (61.11%)		
Type of operation (*n* (%))				
Radical nephrectomy	151 (62.14%)	20 (33.33%)	0.999/0.428	<0.0001
Partial nephrectomy	92 (37.86%)	39 (66.67%)		
Tumor location (*n* (%))				
Right kidney	136 (55.97%)	33 (55.93%)	0.05/<0.01	0.99
Left kidney	107 (44.03%)	26 (44.07%)		
Tumor size, cm (mean ± SD)	5.40 ± 3.12	4.55 ± 3.07	0.93/0.28	0.016 ^d^
Tumor stage (*n* (%))				
pT1	149 (61.31%)	37 (61.11%)	0.481/0.11	0.2753
pT2	20 (8.23%)	9 (11.11%)		
pT3	73 (30.04%)	13 (27.78%)		
pT4	1 (0.40%)	0		
WHO/ISUP grading (*n* (%)) ^a^			0.83/0.17	0.047
G1	94 (38.68%)	11 (26.83%)		
G2	86 (35.39%)	24 (58.54%)		
G3	29 (11.93%)	2 (4.88%)		
G4	34 (13.99%)	4 (9.76%)		
Tumor necrosis area % (mean ± SD)	8.56 ± 18.90	10.25 ± 23.49	0.16/0.085	0.8407
Sarcomatoid area % (mean ± SD)	1.07 ± 5.30	2.46 ± 12.60	0.7/0.204	0.9242 ^d^
Rhabdoid area % (mean ± SD)	0.73 ± 4.42	0	-	-
Lymphatic invasion present (*n* (%))	8 (3.29%)	4 (6.78%)	0.064/0.02	0.7857
Angioinvasion present (*n* (%))	44 (18.11%)	5 (8.47%)	0.346/0.09	0.1089
Neuroinvasion present (*n* (%))	3 (1.24%)	2 (3.39%)	0.082/0.03	0.5518
Renal fibrous capsule invasion present (*n* (%))	132 (54.32%)	27 (45.76%)	0.06/0.18	0.3004
Perinephric fat invasion present (*n* (%))	38 (15.63%)	10 (16.95 %)	0.05/<0.01	0.9999
Renal sinus fat invasion present (*n* (%)) ^b^	33 (21.85%)	4 (6.78%)	0.05/<0.01	0.998
Renal sinus vascular invasion present (*n* (%)) ^b^	35 (23.33%)	3 (5.08%)	0.1/0.04	0.5888
Dead (*n* (%))	79 (32.51%)	6 (10.17%)	0.83/0.17	0.003176

Legend: RCC–renal cell carcinoma, ^a^ rated only for ccRCC and pRCC, ^b^ rated only for cases treated by radical nephrectomy, ^c^
*t*-test, ^d^ Chi^2^ test. SD—standard deviation.

**Table 3 cancers-15-01202-t003:** Comparison of CD44, MMP-2 and MMP-9 immunoreactivity of clear-cell and non-clear-cell renal cell carcinoma cases.

	Overall Immunohistochemical Score	Clear Cell RCC	Non-Clear Cell RCC	Power/Effect Size	*p*-Value
CD44	Group 1 (low expression) (*n* (%))	163 (67.08%)	37 (62.71%)	0.07/0.03	0.8021
	Group 2 (moderate expression) (*n* (%))	49 (20.16%)	13 (22.03%)		
	Group 3 (high expression) (*n* (%))	31 (12.76%)	9 (15.26%)		
MMP-2	Group 1 (low expression) (*n* (%))	240 (98.77%)	34 (57.63%)	0.9/0.56	<0.0001
	Group 2 (moderate expression) (*n* (%))	3 (1.23%)	16 (27.11%)		
	Group 3 (high expression) (*n* (%))	0 (0.00%)	9 (15.26%)		
MMP-9	Group 1 (low expression) (*n* (%))	235 (96.71%)	47 (79.66%)	0.99/0.27	<0.0001
	Group 2 (moderate expression) (*n* (%))	6 (2.47%)	8 (13.56%)		
	Group 3 (high expression) (*n* (%))	2 (0.82%)	4 (6.78%)		

Legend: RCC––renal cell carcinoma. SD—standard deviation.

**Table 4 cancers-15-01202-t004:** Prognostic value of CD44, MMP-2, and MMP-9 expression levels and histopathological subtype of renal cell carcinoma for clinical outcome.

	Coefficient	HR (95%CI)	*p*-Value
CD44 overall immunohistochemical score	0.61	1.8 (1.4–2.4)	0.0000057
MMP-2 overall immunohistochemical score	−1.9	0.15 (0.023–0.990)	0.049
MMP-9 overall immunohistochemical score	0.12	1.1 (0.6–2.1)	0.7
Histopathological subtype–clear cell renal cell carcinoma	0.94	2.6 (1.11–5.88)	0.028

Legend: CI—confidence interval, HR—hazard ratio.

**Table 5 cancers-15-01202-t005:** Multivariate analysis of the relationship between CD44, MMP-2, and MMP-9 expression levels, histopathological subtype of renal cell carcinoma, and overall survival.

	Coefficient	HR (95%CI)	*p*-Value
Histopathological subtype–clear-cell renal cell carcinoma	0.9802	2.6651 (1.1583–6.1312)	0.0211
CD44 overall immunohistochemical score	0.6214	1.8616 (1.4308–2.4222)	0.00000371

Legend: CI—confidence interval, HR—hazard ratio.

## Data Availability

The data presented in this study are available in this article.

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
