# Peer review of "Does the Immunohistochemical Expression of CD44, MMP-2, and MMP-9 in Association with the Histopathological Subtype of Renal Cell Carcinoma Affect the Survival of Patients with Renal Cancer?"

_cancers, 2023, doi:10.3390/cancers15041202_

Round 1

Reviewer 1 Report

The article titled “Does the immunohistochemical expression of CD44, MMP-2, and MMP-9 in association with the histopathological subtype of renal cell carcinoma affect the survival of patients with renal cancer?” is well written and presents interesting results that highlight CD44 as a potential predictive factor for the outcome of RCC patients. Methods and statistical analyses used in this study were adequately applied. The results are sufficiently presented and conclusions drawn adequately follow the results. Because of the use of the immunohistochemistry method for finding new predictor factors and potential therapeutic targets for RCC (CD44) as well as highlighting the need to further study similar molecules within the context of RCC, I propose this article to be accepted for publication after minor revision. Necessary corrections are listed in the following text.

Major comment:

The introduction is well written but does not mention if previous studies that recorded changes of CD44, MMP-2 and MMP-9 expression were recorded/conducted using immunohistochemistry or similar methods in kidney associated cancer types. This should be clearly stated at the end of the introduction before stating the aim of the study because it outlines the knowledge gap that is being covered with this study. Also, please highlight the difference in approach of this and other previously conducted studies that analyzed CD44, MMP-2 and MMP-9.

Minor comments:

Line 85: correct this line (according was written twice), it needs slight rephrasing.

Line 97: Please finish this sentence…were reviewed by two independent researchers/examiners/pathologists.

Line 123: A word is missing in this sentence…All analyses were performed using R Language…please add2

Line 148: In Table 2 please replace the term “Gender” with term “Sex” because Sex is a biological attribute and Gender is a socially constructed attribute.

Line 152: Delete “a” from this sentence… however 157 significant differences…

Line 169: Delete “a” from this sentence…During this time death…

Line 171: Delete “the” from this sentence…All deaths were observed during the 85…

Line 212: Please consider this minor correction…binds to the extracellular matrix and is implicated in cancer cell migration, invasion and metastasis.

Line 229: Please correct this sentence (replace “The” with “A”)…A similar observation…

Line 233: Please correct “wad” in “was”…CD44 expression was an independent risk…

Line 243-246: Please consider splitting this sentence in two parts/two separated sentences for easier reading.

Line 257: Please correct “thee” in “these”

Line 261: Please consider modifying the following sentence because the study indicates its use in predicting the patient outcome and then secondly presents it as a possible therapeutic target (which warrants further investigation)… It may be useful in the search for new therapeutic methods and predicting prognosis of patients with RCC…into…Thus, this finding indicates CD44 as a possible predicting factor for the outcome of RCC patients and possible target for new therapeutic approaches.

Author Response

Dear Reviewer,

Thank You for Your comment and suggestion. We tried to respond to them as best as we could to make our manuscript better quality.

The article titled “Does the immunohistochemical expression of CD44, MMP-2, and MMP-9 in association with the histopathological subtype of renal cell carcinoma affect the survival of patients with renal cancer?” is well written and presents interesting results that highlight CD44 as a potential predictive factor for the outcome of RCC patients. Methods and statistical analyses used in this study were adequately applied. The results are sufficiently presented and conclusions drawn adequately follow the results. Because of the use of the immunohistochemistry method for finding new predictor factors and potential therapeutic targets for RCC (CD44) as well as highlighting the need to further study similar molecules within the context of RCC, I propose this article to be accepted for publication after minor revision. Necessary corrections are listed in the following text.

Major comment:

Point 1: The introduction is well written but does not mention if previous studies that recorded changes of CD44, MMP-2 and MMP-9 expression were recorded/conducted using immunohistochemistry or similar methods in kidney associated cancer types. This should be clearly stated at the end of the introduction before stating the aim of the study because it outlines the knowledge gap that is being covered with this study. Also, please highlight the difference in approach of this and other previously conducted studies that analyzed CD44, MMP-2 and MMP-9.

Response 1: Thank You for Your advice. We have added  this information at the end of the “Introduction” section:

“Till now, there are limited number of prior research investigated the immunohistochemical expression of CD44, MMP-2, and MMP-9 in RCC. The vast majority of studies have explored these prognostic markers only in the context of ccRCC. Moreover not all of the research were recorded using immunohistochemistry, some of them used other techniques. To our best knowledge, it is the largest study investigating prognostic role of CD44, MMP-2, and MMP-9 in different histopathological subtypes of renal cancer.”

Minor comments:

Point 2: Line 85: correct this line (according was written twice), it needs slight rephrasing.

Response 2: We have corrected and rephrased this sentence.

Point 3: Line 97: Please finish this sentence…were reviewed by two independent researchers/examiners/pathologists.

Response 3: We have corrected this sentence.

Point 4: Line 123: A word is missing in this sentence…All analyses were performed using R Language…please add2

Response 4: We have corrected it.

Point 5: Line 148: In Table 2 please replace the term “Gender” with term “Sex” because Sex is a biological attribute and Gender is a socially constructed attribute .

Response 5: We have corrected it and replace “Gender” with term “Sex”.

Point 6: Line 152: Delete “a” from this sentence… however 157 significant differences…

Response 6: We have corrected it and delete “a” from this sentence..

Point 7: Line 169: Delete “a” from this sentence…During this time death…

Response 7: We have corrected it and delete “a” from this sentence.

Point 8: Line 171: Delete “the” from this sentence…All deaths were observed during the 85…

Response 8: We have corrected it and delete “the” from this sentence.

Point 9: Line 212: Please consider this minor correction…binds to the extracellular matrix and is implicated in cancer cell migration, invasion and metastasis.

Response 9: We have corrected it.

Point 10: Line 229: Please correct this sentence (replace “The” with “A”)…A similar observation…

Response 10: We have corrected it.

Point 11: Line 233: Please correct “wad” in “was”…CD44 expression was an independent risk…

Response 11: We have corrected it.

Point 12: Line 243-246: Please consider splitting this sentence in two parts/two separated sentences for easier reading.

Response 12: We have separated this sentence in two sentences.

Point 13: Line 257: Please correct “thee” in “these”

Response 13: We have corrected it.

Point 14: Line 261: Please consider modifying the following sentence because the study indicates its use in predicting the patient outcome and then secondly presents it as a possible therapeutic target (which warrants further investigation)… It may be useful in the search for new therapeutic methods and predicting prognosis of patients with RCC…into…Thus, this finding indicates CD44 as a possible predicting factor for the outcome of RCC patients and possible target for new therapeutic approaches.

Response 14: Thank You for this suggestion. We have modified this sentence.

Reviewer 2 Report

In the present study the authors analyzed whether the immunohistochemical expression of CD44, MMP-2, and MMP-9 in association with the histopathological subtype of RCC affects the survival of patients with renal cancer. 302 patients with diagnosis of RCC, including 243 ccRCC and 59 non- 89 ccRCC (41 pRCC and 18 chRCC) were inculed in study. They found increased CD44 expression was an independent predictor of shorter OS only in the ccRCC group

1.       Were any patients on chemotherapy and if that changed the expression of CD44, MMP-2, MMP-9?

2.       Can the authors include a section on limitations of study, as this is a limited number of patient data?

Author Response

Dear Reviewer,

Thank You for Your comment and suggestion. We tried to respond to them as best as we could to make our manuscript better quality.

In the present study the authors analyzed whether the immunohistochemical expression of CD44, MMP-2, and MMP-9 in association with the histopathological subtype of RCC affects the survival of patients with renal cancer. 302 patients with diagnosis of RCC, including 243 ccRCC and 59 non- 89 ccRCC (41 pRCC and 18 chRCC) were inculed in study. They found increased CD44 expression was an independent predictor of shorter OS only in the ccRCC group

  1. Were any patients on chemotherapy and if that changed the expression of CD44, MMP-2, MMP-9?

Thank You for very interesting question. No, none of the patient included in this study were on chemotherapy. However it is very inquiring to investigate this aspect and check if the exposure to chemotherapy change the expression of CD44, MMP-2, MMP-9 in the tumor cells. We will consider addressing this topic in the future.

  1. Can the authors include a section on limitations of study, as this is a limited number of patient data?

We have added the “Limitation” section after the “Discussion” section:

“The essential limitations of our study were the relatively small number of non-ccRCC tumors included in this study and a large difference in the size of the ccRCC and non-ccRCC groups. It was due to the rarity of papillary and chromophobe RCC when compared to ccRCC. Following limitation concerned the limited number of prior research comparing these two groups of neoplasms. Moreover, there are only few studies describing the role of MMPs in the RCC carcinogenesis, and additionally, not all of them discover these proteins using immunohistochemistry."